# Asatone and Isoasatone A Against *Spodoptera litura* Fab. by Acting on Cytochrome P450 Monoxygenases and Glutathione Transferases

**DOI:** 10.3390/molecules24213940

**Published:** 2019-10-31

**Authors:** Ruimei Ling, Renyue Yang, Ping Li, Xiongfei Zhang, Tunkai Shen, Xiaowen Li, Quan Yang, Lirong Sun, Jian Yan

**Affiliations:** 1Key Laboratory of Agro-Environment in the Tropics, Ministry of Agriculture and Rural Affairs, South China Agricultural University, Guangzhou 510642, China; ruimeiling0725@163.com (R.L.); renyueyang123@126.com (R.Y.); liping2016@scau.edu.cn (P.L.); zhangfeiswymail@163.com (X.Z.); 2Guangdong Provincial Key Laboratory of Eco-Circular Agriculture, South China Agricultural University, Guangzhou 510642, China; 3Guangdong Engineering Research Centre for Modern Eco-Agriculture, South China Agricultural University, Guangzhou 510642, China; 4College of Natural Resources and Environment, South China Agricultural University, Guangzhou 510642, China; 5Key Laboratory of State Administration of Traditional Chinese Medicine for Production & Development of Cantonese Medicinal Materials, School of Traditional Chinese Medicine, Guangdong Pharmaceutical University, Guangzhou 510006, China; shentkcixi@163.com; 6School of Basic Medical Sciences and Key Laboratory of Mental Health of the Ministry of Education, Department of Neurobiology, Southern Medical University, Guangzhou 510642, China; lixw@smu.edu.cn

**Keywords:** induced defenses, anti-insect, *Asarum ichangense*, asatone, isoasatone A, *Spodoptera litura*

## Abstract

Asatone and isoasatone A from *Asarum ichangense* Cheng were determined to be defensive compounds to some insects in a previous investigation. However, the anti-insect activity mechanisms to caterpillar are still unclear. The compounds asatone and isoasatone A from *A. ichangense* were induced by *Spodoptera litura*. The anti-insect activity of asatone and isoasatone A to *S. litura* was further tested by weight growth rate of the insect through a diet experiment. Isoasatone A showed a more significant inhibitory effect on *S. litura* than asatone on the second day. The concentration of asatone was higher than isoasatone A in the second instar larvae of *S. litura* after 12 h on the feeding test diet. Both compounds caused mid-gut structural deformation and tissue decay as determined by mid-gut histopathology of *S. litura*. Furthermore, some detoxification enzyme activity were measured by relative expression levels of genes using a qPCR detecting system. Asatone inhibited the gene expression of the cytochrome P450 monooxygenases (P450s) CYP6AB14. Isoasatone A inhibited the relative expression levels of CYP321B1, CYP321A7, CYP6B47, CYP6AB14, and CYP9A39. Asatone increased the relative gene expression of the glutathione transferases (GSTs) SIGSTe1 and SIGSTo1, in contrast, isoasatone A decreased the relative gene expression of SIGSTe1 by about 33 fold. Neither compound showed an effect on acetylcholinesterase SIAce1 and SIAce2. The mechanism of anti-insect activity by both compounds could be explained by the inhibition of enzymes P450s and GSTs. The results provide new insights into the function of unique secondary metabolites asatone and isoasatone A in genus *Asarum*, and a new understanding of why *A. ichangense* is largely free of insect pests.

## 1. Introduction

Most plants have evolved a range of defense systems to protect themselves when they are infected by insects. Constitutive and inducible secondary metabolites may increase after being fed upon by insects to prevent further damage [1]. Metabolites that were induced in damaged plants could prove useful for preventing and controlling insects. Some extracts from plants can inhibit growth of *Spodoptera litura* Fab. larvae or kill them [2,3,4].

*Asarum ichangense* is a species of the genus *Asarum*. Yi et al. reported that all parts of *A. sieboldii* caused 100% mortality in both *Lycoriella ingenua* and *Coboldia fuscipes* larvae [5]. Essential oil from *A. sieboldii* was toxic to the house dust mite *Dermatophagoide sfarina* [4]. It was determined that a type of neolignane asatone and isoasatone was reported from the genus *Asarum* [6,7,8], and that asatone has proved to be a potent antifeedant to papilionid butterfly, *Luehdorfia puziloi* larvae [9]. Asatone and isoasatone A have seldom been investigated for its toxicity to caterpillars, such as *S. litura*. *S. litura* is a harmful insect which is widespread in many countries, as it causes significant damage to crops every year. It feeds on 181 plant species that belong to 39 families, which include maize (*Zea mays*), sunflower (*Helianthus annuus*), sorghum (*Sorghum bicolor*), and chickpea (*Cicer arietinum*) [10].

Many secondary metabolites demonstrate a number of different mechanisms to control pests. For example, quercetin negatively affected the mid-gut of *S. litura* from *Euphorbia hirta* L [11], and flavonoids, ginkgolide, and bilobalide from *Ginkgo biloba* inhibit the metabolic enzymes in *Hyphantria cunea* larvae by glutathione transferases (GSTs), acetylcholinesterase (AChE), carboxylesterase (CarE), and mixed-functional oxidase (MFO) [12]. Insect cytochrome P450 monooxygenases (P450s) are well-known for their roles in the oxidative metabolism of xenobiotics such as insecticides and plant allelochemicals in pest insects. They have been divided into four clades: CYP2, CYP3, CYP4, and the mitochondrial CYP clade. The clade CYP3 is further subdivided into different CYP families. The CYP321A, CYP6B, and CYP9A subfamilies play a role in the metabolism of xenobiotics and insecticide resistance [13,14,15]. The products of P450-mediated reactions (phase I reactions) can be metabolized by phase II enzymes, such as GSTs, which lead to the production of hydrophilic compounds that can be excreted by phase III transporters [16].

Here, we report that asatone and isoasatone A analyzed from *A. ichangense* can be potential defensive compounds against *S. litura*. The anti-insect mechanisms of both compounds are described using mid-gut histopathology, P450s, GSTs, and AChE of *S. litura*.

## 2. Materials and Methods

### 2.1. Plant and Insect Material

*A. ichangense* was identified by Tieyao Tu, at the South China Botanical Garden, Chinese Academy of Science, and voucher specimens (SCAU20170439) were archived at South China Agricultural University. Dry material of *A. ichangense* was cut into pieces, and liquids were extracted by 95% ethanol three times; the extraction liquids were combined and concentrated to obtain a crude extract under reduced pressure, the yield of ethanol extract of dry *A. ichangense* was about 10%. The standard asatone and isoasatone A were purchased from Fushun Longsheng Biotechnology Co. LTD. *S. litura* pupae were obtained from the Entomology Institute of Sun Yat-sen University, Guangzhou, China. The larvae were reared in an insectary feeding with a normal artificial diet. The insectaries were placed in a room with 25 ± 2 °C room temperature and 70 ± 5% relative humidity and a light:dark photoperiod of 14:10 h. The normal artificial diet consisted of four parts: (A) Kinako 50 g, wheat bran 40 g, yeast powder 13 g, casein 4 g, vitamin C 4 g, and water 250 mL; (B) choline chloride 1 g, sorbic acid 1 g, cholesterol 0.1 g, phaseomannite 0.1 g, and water 50 mL.; (C) agar 13 g and water 250 mL.; and (D) p-hydroxybenzoic acid 1 g. The four parts were mixed well under heat condition, then allowed to cool down to be artificial diet under room temperature. The diet can be stored at 4 °C for one month.

### 2.2. Insect-Induced Plant

*A. ichangense* in mature stage were harvested in the field, then each plant was cultured one pot in greenhouse to adapt to new condition and healthy growth for one month. Then two third-instar larvae were put on one host plant *A. ichangense*. The plant was covered up by a plastic cup to prevent the insects from escaping. One plant different tissues were collected as a biological replicate on the second day for UPLC-MS analysis. The blank without insect was used as control. Three biological replicates plants were performed.

### 2.3. Anti-Insect Assay

The assay to determine activity of asatone and isoasatone A was carried out as following. Test diets at concentrations of 5, 1, and 0 mg/g (artificial diet only) were formulated. Thirty third-instar *S. litura* larvae (body weight 20–30 mg) were each put into a separate transparent plastic box, recorded the corresponding body weight at the same time every day, and fed with a test diet or a normal artificial diet as the control, and they were weighed daily for 3 d. Weight growth rate was calculated relative to the first day *S. litura* larvae weight. Difference in larvae growth between groups was analyzed with one-way analysis of variance (ANOVA) followed by Tukey–Kramer post-hoc tests.

### 2.4. Leaf and Insect Sample Preparation for MS Analysis

Crude extracts of roots, stems, and leaves were dissolved with HPLC grade methanol (Fisher Scientific, Waltham, MA, USA) to a concentration of 1 mg/mL. Insects were ground after being frozen in liquid nitrogen, ground tissue was extracted with methanol (1 mL/300 mg of fresh tissue), extracts were centrifuged at 18,000× *g* for 10 min at 25 °C, and the supernatants were saved for analysis.

Asatone and isoasatone A were analyzed by UPLC-QTOD-MS (Waters, Milford, MA, USA) at a concentration of 5 ppm. UPLC/MS/MS analyses were carried out in an ACQUITY™ UHPLC system (Waters, Milford, MA, USA) with cooling auto-sampler and column oven that enabled to control temperature. An ACQUITY UPLC^®^ BEH C18 column (2.1 mm × 50 mm, 1.7 μm) was employed, and the column temperature was maintained at 40 °C. The solvent was acetonitrile contained 0.1% formic acid as mobile phase system A, water contained 0.1% formic acid for another mobile phase system B. The following program for a linear elution gradient was applied: 0–1 min, 40% A, 1–2 min, 30% A, 2–3 min, 10% A, 3–4 min, 10% A, 4–5 min, 5% A, 5–6 min, 40% A, and 6–7 min, 40% A. The flow rate was set at 0.4 mL/min. The auto-sampler was conditioned at 23 °C, and the injection volume was 2 μL for analysis.

Mass spectrometric detection was performed on a Xevo TQD equipped with an electrospray ionization source (ESI). The capillary voltage was set to 3.30 kV, and the source temperature was maintained at 150 °C. N_2_ gas was used as desolvation at a temperature of 300 °C and Ar gas for collision. The desolvation and cone gas flow rates were 800 and 50 L/h, respectively.

The two compounds were optimized for multiple reaction monitoring using positive mode. For the selection of detected ions, *m*/*z* 449.3058 was selected as the parent ion under the 16 V cone voltage, while *m*/*z* 193.0690 and 385.2202 were set as the daughter qualitative and quantitative ions, respectively, with the corresponding collision energy settings at 28 V and 10 V, respectively. The dwell time was 0.52 s.

### 2.5. Expression Levels of Different Genes in Different Tissues

Test diets at concentrations of 0 or 1 mg/g were fed to 30 fifth-instar larvae for 2 d, then head and mid-gut tissues of *S. litura* were dissected on the second day after storing those samples in liquid nitrogen. Total *S. litura* RNA was isolated with Trizol according to the manufacturer’s protocol (Invitrogen/ThermoFisher Scientific, Waltham, MA, USA). The purity and quantity of RNA was assessed using a NanoDrop ND2000 (NanoDrop Technologies, Wilmington, DE, USA) and monitored on 1% agarose gels, respectively. A 1 μg sample of total RNA was prepared to synthesize single-stranded cDNA with a ThermoScript™ RT-PCR system kit (Thermo Fisher Scientific, Waltham, MA, USA) following the manufacturer’s instructions, then cDNA was diluted 1:10 in water for RT-qPCR. The relative expression levels of all genes were quantified by RT-qPCR and reactions were performed on the DNA Engine Opticon 2 Continuous Fluorescence Detection System (MJ Research Inc., Waltham, MA, USA) using gene specific primers (see Appendix A). The reaction volume of all samples was 20 μL (10 μL SYBR qPCR Mix, 0.4 μL forward primer, 0.4 μL reverse primer, 5 μL cDNA, and 4.2 μL H_2_O). β-actin was used to normalize the transcript abundance among tested samples. We used a SYBR Green I Master Mix (Roche Diagnostics Corp., Indianapolis, IN, USA) with the following thermal program: 95 °C for 10 s, followed by 40 cycles at 95 °C for 10 s, 60 °C for 20 s. The homogeneity of the PCR product was confirmed by a melting curve analysis. The ratios of the target gene/β-actin gene values were calculated according to the 2^−ΔΔCt^ method. In the experiment, three *S. litura* were put together as one biological replicate, and five biological replicates of target genes were analyzed in triplicate by RT-qPCR.

### 2.6. Histopathology Study

Histopathology of the mid-gut was based on a published method with minor modifications [11,17]. All of larvae with or without treatment were cut through the cuticle longitudinally and fixed in 4% formaldehyde. The dehydration of tissues was done consecutively with formaldehyde for 30 min, with 80% methanol for 30 min, with 90% methanol for 30 min, with 95% methanol for 90 min, and with 100% methanol for 60 min. After dehydration, the samples were placed in xylene for 40 min followed by embedding for about 80 min in a warm oven with wax. Then, the fluid wax with the sample was poured in paper boats and cooled to prepare wax blocks, which were then sliced by a microtome at 2 μm per section. Sectioned tissues were de-waxed by xylene (100%) for 8 min. To achieve the transparency of the slice, rehydration of sectioned tissues was made sequentially with 100%, 95%, and 90% alcohol, and then with distilled water for 30 s. Tissues were stained in Ehrlich’s hematoxylin for 10–15 min, placed in distilled water for 20 s, differentiated in 1% hydrochloric acid alcohol for 10 s, placed in distilled water for 15 min, and stained in 1% eosin for 3 min. Then, dehydration was conducted sequentially again with 90%, 95%, 95%, and 100% alcohol for 30 s, tissues were dipped in xylene twice, and then mounted with one drop DPX (distyrene plasticizer xylene). The observations of mid-gut cells of *S. litura* were photographed by a OLYMPUS DP7 microscope connected to a computer. The actual sites of action in the mid-gut of treated larvae were observed and compared with the control. One *S. litura* was used as a biological replicate, and three biological replicates were performed.

## 3. Results

To investigate the potential active constituent of *A. ichangense*, two peaks induced were detected using UPLC-MS by *S. litura* feeding. The two peaks were identified as asatone and isoasatone A using standards, respectively (Figure 1). Concentrations of asatone and isoasatone A were much higher in roots than in stems or leaves from fresh plants, and concentration of asatone was 21.3 times higher than isoasatone A in the roots (Figure 1C,D). Isoasatone A can be significantly induced in the roots, stems, and leaves by *S. litura* feeding; asatone also can be induced abundantly in stems and leaves, however it did not vary in roots (Figure 1C,D).

In the assay for screening active concentrations of asatone and isoasatone A, the concentration of 1 mg/g and were detected in the body of *S. litura* (Figure 2D).

In the mid-gut histopathology, the mid-gut of *S. litura* displayed a complete layer of epithelial cells and columnar cells, and an epithelial layer was noticeable in the control (Figure 3A). However, in the treatment group with both compounds, the mid-gut of *S. litura* showed some morphological and cellular damage of epithelial columnar cells, and the epithelial cells were enlarged and appeared disorganized, marked with green arrow (Figure 3B,C).

Isoasatone A exhibited down-regulation of the gene expression of *P450 CYP321B1*, *CYP321A7*, *CYP6B47*, *CYP6AB14*, and *CYP9A39*, but not *CYP6B58* (Figure 4A–F). Asatone obviously showed down-regulation of the gene expression of *P450 CYP6AB14* (Figure 4D). The relative gene expression of *SIGSTe1* and *SIGSTo1* can be increased by feeding artificial diet containing asatone, meanwhile *SIGSTo1* also can be increased by feeding artificial diet containing isoasatone, but, in contrast, isoasatone A extremely decreased relative gene expression of *SIGSTe1* for about 33 times compared with control (Figure 5A,B). However, both compounds showed no effect on *SIAce1* and *SIAce2* (Figure 6A,B). Inhibition on detoxification enzymes GSTs and P450 explained partly the mechanism of anti-insect activity.

## 4. Discussion

Asatone and isoasatone A belonging to neolignanes type chemical structure are characteristic constituent in genus *Asarum* [18]. The concentration of asatone was much higher than isoasatone A in roots, leaves, and stems of *A. ichangense* in normal condition (Figure 1C,D). Most previously insect active investigation mainly focused on butterfly [9]. However, our results showed that both compounds caused *S. litura* to lose weight (Figure 2A–C), which supported the mid-gut histopathology. Ingestion of isoasatone A and asatone caused marked changes in the midgut of *S. litura* relative control, such as cell swelling, and lysis (Figure 3A–C). Meanwhile, in fresh plants feeding by *S. litura* induced to synthesize more isoasatone A in the roots, stems, and leaves, but asatone was a little bit induced in stems and leaves only (Figure 1C,D). We further hypothesize that isoasatone A, an isomer of asatone, is more helpful to protect *A. ichangense* from *S. litura* or other insects than asatone. This was consistent with the action of glutathione transferase, where relative gene expression of SIGSTe1 was extremely lower for isoasatone (Figure 6A).

P450s play crucial roles in the metabolism of endogenous and dietary compounds in insects [19]. Generally, asatone and isoasatone A inhibit gene expression level of *P450* genes, except for *CYP6B58*, and this explained why both compounds inhibited the growth of *S. litura* (Figure 4A–F). Based on phylogenetic analysis, *CYP6B47* and *CYP6B58* are very similar to the P450s in *S. litura* (Appendix A), but asatone and isoasatone A had the opposite effect on CYP6B47 and CYP6B58 (Figure 4C,E). Therefore, additional reasons need to be investigated further to explain this paradox.

Glutathione transferases are another family of enzymes P450s related to the detoxification of endogenous metabolites and exogenous chemicals [20]. Gene expression levels of *SlGSTe1* and *SlGSTo1* (Figure 5A,B) increased after larvae ingested asatone, which was similar to previous reports on the xenobiotic compounds xanthotoxin and chlorpyrifos [21]. However, isoasatone A inhibited the expression levels of SlGSTe1 dramatically (Figure 5A,B), which suggested that isoasatone A was toxic to *S. litura*. Moreover, we detected that isoasatone A is more consumed and absorbed by the insect itself than asatone (Figure 2D). Maybe, isoasatone A can be transfer to other toxic compound to act on the insect, more metabolite process is worthy of carrying out in the future work.

In conclusion, the mechanism of anti-insect activity by both compounds could be explained by inhibition of enzyme P450s and GSTs based on target gene expression level. The results provide new insights into the function of unique secondary metabolites asatone and isoasatone A in genus *Asarum* and a new understanding of why *A. ichangense* is largely free of insect pests.

## Figures and Tables

**Figure 1 molecules-24-03940-f001:**
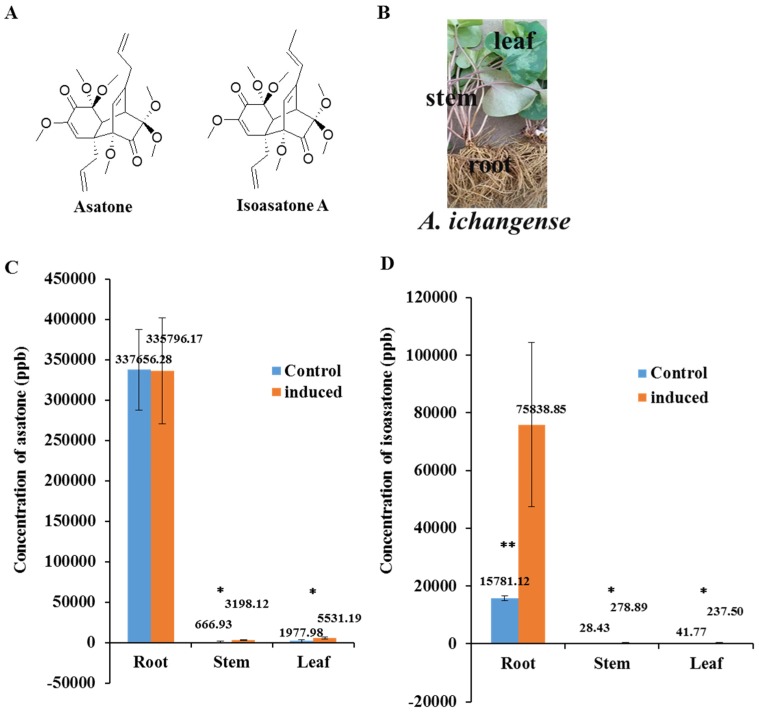
The compounds asatone and isoasatone A distribution in *Asarum ichangense*. (**A**) Chemical structures asatone and isoasatone from *A. ichangense*. (**B**) Photos showing *A. ichangense*. (**C**) Concentration of asatone with (induced) and without (control) *S. litura* feeding on roots, stems, or leaves. (**D**) Concentration of isoasatone A with (induced) and without (control) *S. litura* feeding. *e.* mean +/− s.e.; *n* = 3; * *p* < 0.05, *t*-test.

**Figure 2 molecules-24-03940-f002:**
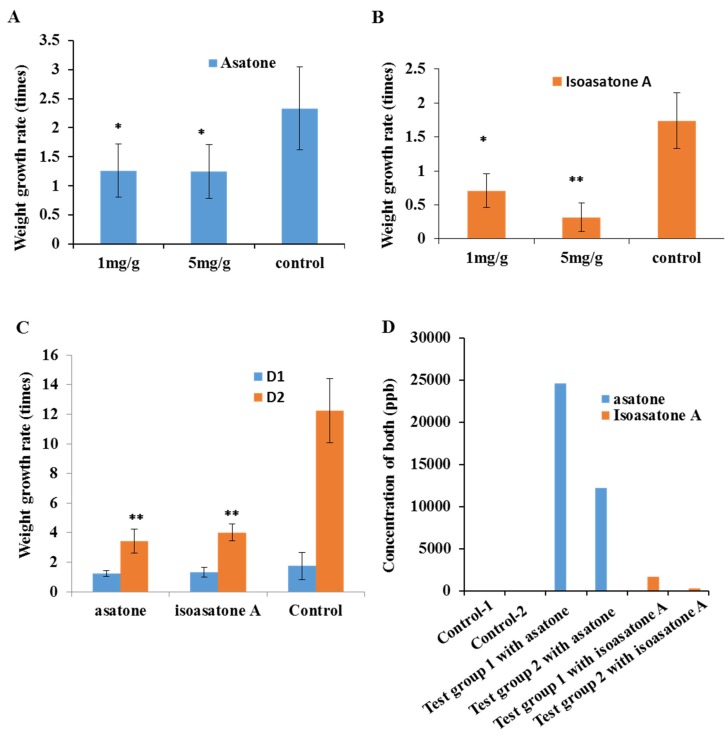
Anti-insect activity of asatone and isoasatone A. (**A**) Different concentrations of asatone against *Spodoptera litura* after 2 d. (**B**) Different concentrations of isoasatone A against *Spodoptera litura* after 2 d. (**C**) Growth rate of *Spodoptera litura* growth rate on an artificial diet (1 mg/g) with different day. (**D**) Concentration of asatone and isoasatone A in *S. litura* by feeding test artificial diet. Growth rate with one asterisk indicates a significant difference (*p* < 0.05) and a double asterisk indicates significance at *p* < 0.01 relative to control in a *t*-test.

**Figure 3 molecules-24-03940-f003:**
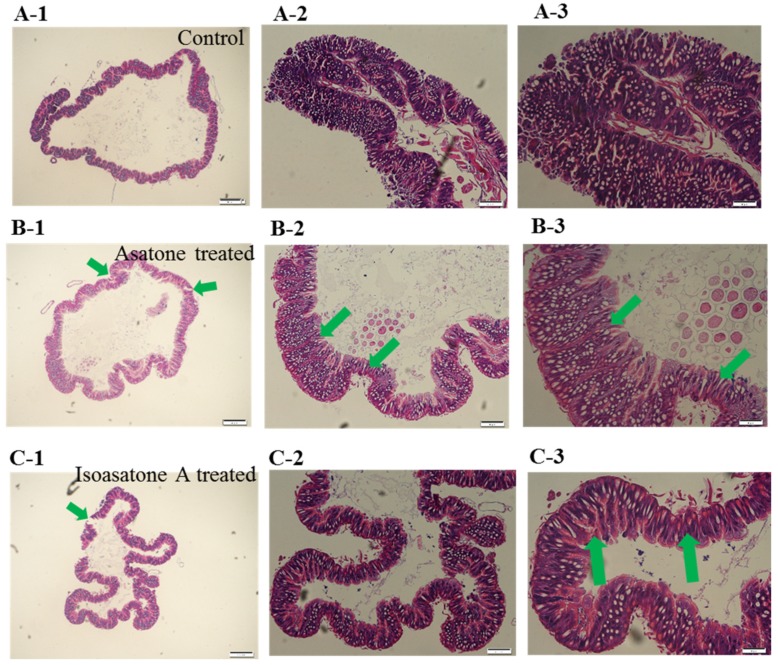
Midgut histopathology of *Spodoptera litura* after treatment with each compound asatone and isoasatone A compared with control. (**A**) Control feeding diet only, (**B**) test group on asatone diet, and (**C**) test group on isoasatone A diet. Note: label 1,2,3 present different magnification.

**Figure 4 molecules-24-03940-f004:**
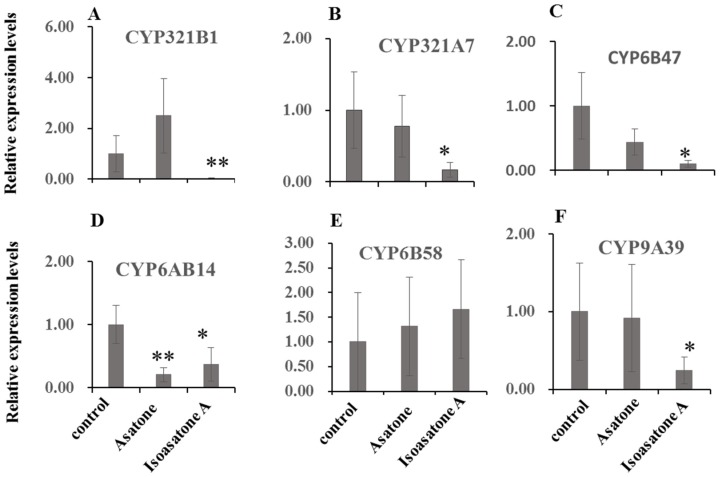
Expression levels of selected genes. (**A**–**F**) P450s, β-actin gene served as an internal reference to determine relative expression levels. Data shown are mean +/− s.e.; *n* = 3–5. Asterisks indicate significant differences relative to control (* *p* < 0.05, ** *p* < 0.01) in a *t*-test.

**Figure 5 molecules-24-03940-f005:**
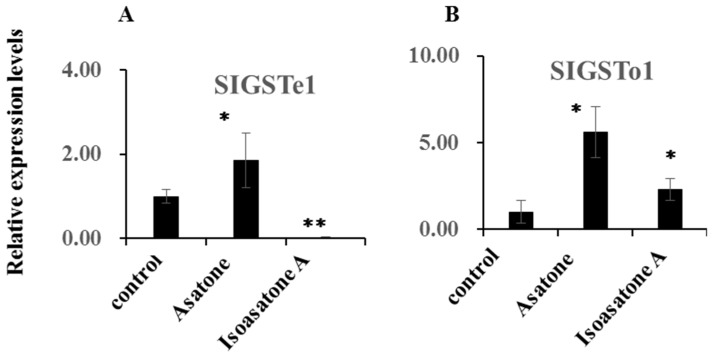
Expression levels of selected genes. (**A**,**B**) GSTs, β-actin gene served as an internal reference to determine relative expression levels. Data shown are mean +/− s.e.; *n* = 3–5. Asterisks indicate significant differences relative to control (* *p* < 0.05, ** *p* < 0.01) in a *t*-test.

**Figure 6 molecules-24-03940-f006:**
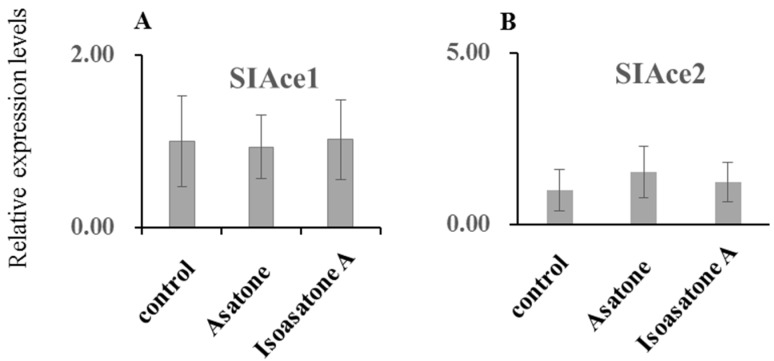
Expression levels of selected genes. (**A**,**B**) acetylcholinesterase. The β-actin gene served as an internal reference to determine relative expression levels. Data shown are mean +/− s.e.; *n* = 3–5. Asterisks indicate significant differences relative to control (* *p* < 0.05, ** *p* < 0.01) in a *t*-test.

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
