# Peer review of "Asatone and Isoasatone A Against Spodoptera litura Fab. by Acting on Cytochrome P450 Monoxygenases and Glutathione Transferases"

_molecules, 2019, doi:10.3390/molecules24213940_

Round 1

Reviewer 1 Report

Manuscript entitled: “Asatone and isoasatone against Spodoptera litura  Fab. by acting on enzyme P450s and glutathione transferases” describe anti-insect activity of defence compounds asatone and isoasatone isolated from plants Asarum ichangense on caterpillars Spodoptera litura Fab. The effect was followed by weight of larvae feed on these two compounds, by mid-gut histopathology and expression level of some detoxification enzymes like cytochromes P450, glutathione transferases or acetylcholinesterase. The topic is of interest, there is a constant search for new anti-insect compounds.

Although I’m not an English speaking reviewer I would like to suggest authors to edit the manuscript by a native speaking professional first to increase the intelligibility.

I would as well suggest verifying guidelines for authors about the manuscript format. For example: no explanation of abbreviation in the abstract, different size of font, non-italic font with Latin names, different paragraph spacing, and inconsistent capitalisation of text in figures. No Author Contributions and Funding is listed.

The text under the Figure 4 and Figure 5 is the same.

Methods are undercitied, type of balance should be described more in details.

The template to design primers is not clear; Suppl. table 1 text should fit in lines. Similarity of P450 in phylogenetic analysis of P450 by NJ method doesn’t tell much about the function of listed P450.

The meaning of the sentence is not clear: “Inhibition on detoxification enzymes GSTs and P450 explained the mechanism of anti-insect activity somewhat.”

The sentence in discussion need citations: “Most previously insect active investigation mainly focused on butterfly.”

Author Response

Manuscript entitled: “Asatone and isoasatone against Spodoptera litura  Fab. by acting on enzyme P450s and glutathione transferases” describe anti-insect activity of defence compounds asatone and isoasatone isolated from plants Asarum ichangense on caterpillars Spodoptera lituraFab. The effect was followed by weight of larvae feed on these two compounds, by mid-gut histopathology and expression level of some detoxification enzymes like cytochromes P450, glutathione transferases or acetylcholinesterase. The topic is of interest, there is a constant search for new anti-insect compounds.

Although I’m not an English speaking reviewer I would like to suggest authors to edit the manuscript by a native speaking professional first to increase the intelligibility.

>> Thanks. I revised and polished the manuscript.

I would as well suggest verifying guidelines for authors about the manuscript format. For example: no explanation of abbreviation in the abstract, different size of font, non-italic font with Latin names, different paragraph spacing, and inconsistent capitalisation of text in figures. No Author Contributions and Funding is listed.

>>We had corrected them, and added the author contribution in the text.

The text under the Figure 4 and Figure 5 is the same.

>> we corrected them.

Methods are undercitied, type of balance should be described more in details.

>>we revised some sentences in method section.

The template to design primers is not clear; Suppl. table 1 text should fit in lines. Similarity of P450 in phylogenetic analysis of P450 by NJ method doesn’t tell much about the function of listed P450.

>>Thanks, we corrected it. Suppl. table 1 text were filled lines.

The meaning of the sentence is not clear: “Inhibition on detoxification enzymes GSTs and P450 explained the mechanism of anti-insect activity somewhat.”

>>Thanks, we changed the sentence as “Inhibition on detoxification enzymes GSTs and P450 explained partly the mechanism of anti-insect activity”

The sentence in discussion need citations: “Most previously insect active investigation mainly focused on butterfly.”

>> ok, We added a reference in the text.

Reviewer 2 Report

Manuscript title: Asatone and isoasatone against Spodoptera litura by acting on enzyme P450s and glutathione transferases

General comments:
The submitted manuscript has been focused in the study that the biosynthesis of ssatone and isoasatone could enhance the inhibition activity on the growth of Spodoptera litura. In general, the authors have completed a reasonable study with very informative data and finding of active mechanisms of asatone and isoasatone on the growth inhibition of the insect, some of the concerned questions would be noted as the following:
1. Two isomers of isoasatone have been reported as isoaasatone A and B, please describe the identification methods for differentiating the structure used in the study.

2. The differences of the ethanol extract from Asarum ichangense and the induced plant should be presented from the UHPLC to indicate the changes of asatone and isoasatone after the induction.

3. Line 166-167. The two peaks were identified as asatone and isoasatone using standard, respectively.
Please indicate the standards source.

4. Line 254 establish should be revised to “established”.

Author Response

General comments:
The submitted manuscript has been focused in the study that the biosynthesis of ssatone and isoasatone could enhance the inhibition activity on the growth of Spodoptera litura. In general, the authors have completed a reasonable study with very informative data and finding of active mechanisms of asatone and isoasatone on the growth inhibition of the insect, some of the concerned questions would be noted as the following: 
1. Two isomers of isoasatone have been reported as isoaasatone A and B, please describe the identification methods for differentiating the structure used in the study.

>>we corrected isoasatone, refer to isoasatone A.

The differences of the ethanol extract from Asarum ichangense and the induced plant should be presented from the UHPLC to indicate the changes of asatone and isoasatone after the induction.

>> We directly detected the content of Asatone and isoasatone A under normal condition and induced condition using UPLC-MS by MRM analysis model, please refer to Fig.1c and 1d.

Line 166-167. The two peaks were identified as asatone and isoasatone using standard, respectively.
Please indicate the standards source.

>> the source was added in the text.

Line 254 establish should be revised to “established”.

>>Thanks, revised it.

Round 2

Reviewer 1 Report

line 3: should be changed into ...Fab. by acting on cytochrome P450 monoxygenases and glutathione...

line 25: change testified into tested

line 31: cytochrome without a capital letter

line 40 & else where: isoasatone A should be written consistently through the text, also in introduction and figure legends

line 82: The larvae were reared in an insectary under the condition with 25  2 °C and 70  5% relative 83 humidity at a light : dark photoperiod of 14:10 h feeding with a normal artificial diet. - refrase this sentence.

Line 91: ... to adapt to new ...

Line 93: Which tissues? does thee biological replicats means 3 plants?

Line 100: How were larvae marked?

Line 105: Crude extracts (not extractions)

Line 170: standards purchased from Fushun Longsheng Biotechnology Co. LTD - move into material section

Line 177 B part of the figure: red color could be replaced wih more visible one

Line 179 & else where: Capitalise word after full stop. ((B) Photos ...

Line 200: 2x full stop

Line 206: remove obviously

Line 211: add for about 33 times

Line 231: change belong to belonging

Line 259: change into ... enzymes P450s ...

Line 263: 2x full stop

Line 273: change into ... assitance in preparation...

Suppl. Table S1: Will it be formated by publisher?

Suppl. Fig. S2: Verify the meaning of the caption: The scale bar indicates 0.2
amino acid substitutions per site. Give credits to the method which was applied.

Author Response

Comments and Suggestions for Authors

line 3: should be changed into ...Fab. by acting on cytochrome P450 monoxygenases and glutathione...

>>it was corrected

line 25: change testified into tested

>>it was corrected

line 31: cytochrome without a capital letter

>>it was corrected

line 40 & else where: isoasatone A should be written consistently through the text, also in introduction and figure legends

>> ok, We checked the whole paper and revised them in introduction line 54 and in Fig2, Suppl. Fig. S1.

line 82: The larvae were reared in an insectary under the condition with 25 ± 2 °C and 70 ± 5% relative 83 humidity at a light : dark photoperiod of 14:10 h feeding with a normal artificial diet. - refrase this sentence.

>>ok, the sentence was rewrote.

Line 91: ... to adapt to new ...

>> It was corrected.

Line 93: Which tissues? does thee biological replicats means 3 plants?

>> it was corrected.

Line 100: How were larvae marked?

>> the sentence was revised as “recorded the corresponding body weight at the…”

Line 105: Crude extracts (not extractions)

>> it was corrected.

Line 170: standards purchased from Fushun Longsheng Biotechnology Co. LTD - move into material section

>> it was moved into the material section

Line 177 B part of the figure: red color could be replaced wih more visible one

>>ok, it was changed.

Line 179 & else where: Capitalise word after full stop. ((B) Photos ...

>> ok, it was revised.

Line 200: 2x full stop

>>it was corrected

Line 206: remove obviously

>> The word was deleted

Line 211: add for about 33 times

>>ok, the word was added in the text.

Line 231: change belong to belonging

>>ok, it was revised in the text.

Line 259: change into ... enzymes P450s ...

>>ok, it was corrected

Line 263: 2x full stop

>>ok, it was corrected

Line 273: change into ... assitance in preparation..

>>ok.

Suppl. Table S1: Will it be formated by publisher?

>>Ok.

Suppl. Fig. S2: Verify the meaning of the caption: The scale bar indicates 0.2
amino acid substitutions per site. Give credits to the method which was applied.

>>Thanks. The legend was rewrote.